Effectiveness of treatment modalities for childhood esotropia: a systematic review

Alrasheed Saif Hassan S.rasheed@qu.edu.sa
Challa Naveen Kumar
Aljohani Saeed sa.aljohani@qu.edu.sa
Almutairi Nawaf M.
Alnawmasi Mohammed M.
Department of Optometry, College of Applied Medical Sciences, Qassim University , Buraydah , Saudi Arabia
Fogt Nick
Electronic publication date: 2025 Jun 16
Publication date: 2025
Volume: 13
Electronic Location ID: e19584
Received 2024 Nov 1; Accepted 2025 May 19
Copyright: ©2025 Alrasheed et al.
Copyright year: 2025
Copyright holder: Alrasheed et al.
License: This is an open access article distributed under the terms of the Creative Commons Attribution License, which permits unrestricted use, distribution, reproduction and adaptation in any medium and for any purpose provided that it is properly attributed. For attribution, the original author(s), title, publication source (PeerJ) and either DOI or URL of the article must be cited.
License URL: https://creativecommons.org/licenses/by/4.0/

Keywords: Child, Infantile Esotropia, Global Health, Botulinum toxin, Follow Up, Symptoms

Funding: Deanship of Graduate Studies and Scientific Research at Qassim University QU-APC-2025 The Deanship of Graduate Studies and Scientific Research at Qassim University provided financial support (QU-APC-2025). The funders had no role in study design, data collection and analysis, decision to publish, or preparation of the manuscript.

==============================
Background

Esotropia has several types that commonly manifest in early childhood, with numerous treatment options described in the literature. The aim of this systematic review is to synthesize recent research findings on the management of childhood esotropia (ET) to clarify the relative success rates and specific indications for each treatment option, providing guidance for eye care professionals in selecting the most effective interventions.

Methodology

A comprehensive search was conducted across multiple databases, including PubMed, Web of Science, ProQuest, Scopus, Google Scholar, EBSCO, and Medline, following PRISMA 2020 guidelines. The search was restricted to articles published between 1990 and 2023 that examined various treatment modalities for different types of childhood esotropia (ET). In this study, success was defined as a post-treatment deviation of less than 10 prism dioptres (PD). The study protocol was registered in the International Prospective Register of Systematic Reviews (PROSPERO) under registration number CRD42024589042.

Results

The final systematic review included 34 studies from 14 countries, encompassing 3,877 children with a mean age of 4.72 ± 2.08 years. The reviewed studies indicated that optical correction had the highest effectiveness rate at 79.31% after an average follow-up of 5.57 years. Extraocular muscle surgery demonstrated an effectiveness rate of 71.4% with a follow-up period of 2.89 years, while botulinum toxin (BT) injections showed a lower effectiveness rate of 61.24% after a follow-up of 3.15 years.

Conclusions

The review concluded that substantial evidence supports full cycloplegic hyperopic correction as the most effective first-line treatment for childhood accommodative esotropia (AET). However, surgery may be required for some children with AET if their deviation remains over 15 PD after full cycloplegic hyperopic correction. Prismatic correction was highly successful in managing residual deviation in partial AET. Bilateral BT injections proved effective as a first-line treatment for acute-onset ET and infantile ET without a vertical component. Timely surgical intervention enhances sensory outcomes in infantile ET; however, no surgical technique has demonstrated a distinct advantage.

Introduction

Esotropia (ET), a type of strabismus, involves an inward turning of the eyes and commonly manifests in early childhood, with a prevalence of approximately 1% in the population (Wright, Spiegel & Thompson, 2006; Issaho et al., 2017). In general, strabismus affects approximately 4% of the population (Wright, Spiegel & Thompson, 2006). If a parent has strabismus, the child’s risk of being affected is four times higher compared to children of parents without strabismus (Alrasheed, Naidoo & Clarke-Farr, 2016). Approximately 60% of affected children with ET have a family history of hyperopic accommodative esotropia, which is a common form of childhood strabismus (Wright, Spiegel & Thompson, 2006; Issaho et al., 2017; Alrasheed, Naidoo & Clarke-Farr, 2016; Cole & Camuglia, 2012; Guo et al., 2022). The condition has different types, which include infantile esotropia (IET), also known as congenital esotropia, which is typically characterized as a significant inward deviation of the eyes that manifests before the age of 6 months, with a reported prevalence at birth of 27 per 10,000 live births (Wright, Spiegel & Thompson, 2006; Issaho et al., 2017). Whereas an earlier study reported that the incidence of IET was 0.5% in a sample of 582 babies (Alrasheed, Naidoo & Clarke-Farr, 2016). IET is commonly unrelated to refractive errors (RE) and is commonly accompanied by additional motor abnormalities, such as inferior oblique over-action, dissociated vertical deviation (DVD), and latent nystagmus (Wright, Spiegel & Thompson, 2006; Cole & Camuglia, 2012). IET can arise in infancy due to various disorders, which include congenital fibrosis of the extraocular muscles, Duane’s syndrome, and infantile myasthenia gravis (Guo et al., 2022). According to a recent study (Brodsky, 2018), IET can be attributed to disruptions in the development of binocularity in the striate cortex during the first three months of life. These disruptions lead to the re-activation of subcortical optokinetic pathways primarily driven by one eye, causing a marked nasal-ward bias in the early stages and resulting in the inward positioning of the eyes.

Childhood accommodative esotropia (AET) usually occurs between 2–4 years old. The key to diagnosing this condition is identifying hyperopia greater than 2.00 dioptres (D), typically ranging from 2 to 6 D, along with a variable angle of deviation at onset. (Wright, Spiegel & Thompson, 2006; Basheer, Deyabat & Mocan, 2023). AET is associated with the activation of the accommodation reflex; due to this association, AET can be further classified as refractive accommodative esotropia (RAET), non-refractive accommodative esotropia (NAET), or partially accommodative esotropia(PAET) (Basheer, Deyabat & Mocan, 2023; Alrasheed et al., 2023). AET can manifest at any stage in a child’s life, although it is most common in children between 2 and 4 years old. For children with AET associated with hyperopia, it tends to worsen until they reach 6 to 7 years of age. After that, the severity of the hyperopia decreases until the child turns 14 years old (Ha, Suh & Kim, 2018). Whereas non-refractive accommodative esotropia commonly occurs in children with high accommodative convergence. These children tend to exhibit proper alignment of the eyes when looking at distant objects. However, they experience more than 10 prism dioptres (PD) of esotropia, when focusing on near objects (Alrasheed et al., 2023; Ha, Suh & Kim, 2018). Children diagnosed with non-accommodative esotropia may exhibit a broad spectrum of refractive error (RE), varying from myopia to high hyperopia. When it comes to children with PAET, full RE correction often leads to a decrease in the angle of esotropia. Still, some residual deviation may persist despite receiving treatment for amblyopia and undergoing full hyperopic correction (Alrasheed et al., 2023; Kim & Kim, 2014).

Accommodation is a factor that contributes nearly 50% of childhood esotropia (Aljohani et al., 2022; Robert & Rutstein, 1998). Greenberg et al. (2007) conducted a study to describe the incidence and types of childhood esotropia over 10 years of follow-up. They showed that the prevalence was 2% and more common in children younger than six years; the most common type was full accommodative, followed by acquired non-accommodative. Additionally, Mohney et al. (2011) reported that most children diagnosed with AET need spectacle correction throughout their adolescence. Only a minority of children with full AET may eventually require surgical intervention. There are numerous treatment options for several types of childhood esotropia described in the literature. Hence, this systematic review is mainly focused on discussing the different management options for the treatment of childhood esotropia and their efficacy in improving ocular alignment.

Esotropia is one of the most common types of strabismus in children and can present in various forms, including accommodative, non-accommodative, acute onset, and infantile, with each type requires a specific management approach. Treatment options for childhood ET include optical correction, extraocular muscle surgery, botulinum toxin (BT) injections, and prismatic correction, yet their effectiveness can vary significantly. With advancements in ophthalmologic interventions, new data on the efficacy of treatments like BT injections and prismatic correction have emerged. So the rationale of the present systematic review is to synthesize recent research findings to clarify the relative success rates and specific indications for each treatment option, ultimately guiding the eye care professionals in selecting the most effective interventions.

Methods

Study design

This systematic review was reported in accordance with the guidelines outlined in the Preferred Reporting Items for Systematic Reviews (PRISMA, 2020) (Page et al., 2021). The review protocol was registered in the International Prospective Register of Systematic Reviews (PROSPERO) under registration number CRD42024589042. The researchers conducted a comprehensive search across multiple online databases, including PubMed, Web of Science, ProQuest, Scopus, Google Scholar, EBSCO, and Medline, covering studies published from January 1990 to December 2023. The quality of each study included in this review was assessed using the evaluation tool developed by Downs & Black (1998). Additionally, each selected article underwent a thorough review process and was assigned a score on a scale of 1 to 10, as shown in Table 1. This review included a diverse range of articles to evaluate various treatment approaches for different forms of childhood esotropia, conducted across multiple countries and involving children from diverse age groups.

Table 1 Features of studies evaluated childhood esotropia treatment.

Author and published year	Country	Age group (years)	Age (mean (SD)	Study design	Sample size	Quality assessment score	
Mohney et al. (2011)	USA	2.2–17	4	Retrospective	244	10	
Gerling & Arnoldi (2013)	USA	3–7	–	Retrospective	36	9	
Mohan & Sharma (2014)	India	12	4.81 ± 2.64	Retrospective	107	10	
Whitman, MacNeill & Hunter (2016)	USA	–	5.1 ± 2.1	Retrospective	180	9	
Mezera, Wygnanski-Jaffe & Stolovich (2015)	Israel	–	5.36 ± 2.68	Retrospective	35	9	
Sreelatha et al. (2022)	Oman	2–6	–	Retrospective	51	9	
Iqbal, Hussain & Qazi (2013)	Pakistan	2–15	7.17 ± 2.07	Cohort	44	10	
Mulvihill et al. (2000)	UK	2–9.5	4.2	Cohort	103	10	
Krishnamoorthy (2019)	India	1–4	–	Cohort	23	9	
Choe, Yang & Hwang (2019)	Korea	10	–	Retrospective	124	10	
Kekunnaya, Velez & Pineles (2013)	USA	0.5–2.5	1.3	Retrospective	6	10	
Lambert et al. (2003)	USA	–	4.2 ± 1.5	Prospective	20	9	
Yun-chun & Longqian (2011)	China	–	7	Retrospective	45	9	
Wabulembo & Demer (2012)	USA	–	4.3 ± 1.6	Prospective	21	10	
Akar et al. (2013)	Turkey	–	2.9 ± 1.3	Retrospective	473	9	
Pensiero et al. (2021)	Italy	0–4	–	Cohort	86	10	
Alshamlan et al. (2021)	Saudi Arabia	–	4.7 ± 4.4	Prospective	56	10	
AlShammari, Alaam & Alfreihi (2022)	Saudi Arabia	1–14	–	Retrospective	224	10	
Scott et al. (1990)	USA	2m–12	–	Prospective	413	10	
De Alba Campomanes, Binenbaum & Eguiarte (2010)	USA	3	–	Prospective	442	9	
McNeer, Tucker & Spencer (1997)	USA	0.5–4	–	Prospective	76	9	
Campos, Schiavi & Bellusci (2000)	Italy	2–9	–	Cohort	60	9	
Gursoy et al. (2012)	Turkey	2	–	Retrospective	51	9	
Flores-Reyes et al. (2016)	Mexico	2–12	6.43	Prospective	21	10	
Arnoldi (2002)	USA	–	–	Retrospective	108	9	
Alam et al. (2023)	Saudi Arabia	1–2	1.5± 0.5	Cohort	63	10	
De Alba Campomanes, Binenbaum & Eguiarte (2010)	USA	3	3	Prospective	442	9	
Wan et al. (2017)	USA	–	6	Retrospective	49	8	
Gama et al. (2020)	Portugal	–	6	Retrospective	48	8	
Jiang et al. (2023)	China	1–17	7	Retrospective	52	9	
Shi et al. (2021)	China	1–17	–	Prospective	60	9	
Bayramlar et al. (2014)	USA	1–14	–	Prospective	18	10	
Elkhawaga et al. (2022)	Egypt	1–15	–	Prospective	61	10	
Kim & Choi (2017)	South Korea	31–17	–	Retrospective	35	9	
All			4.72 ± 2.08		3,877	9.38	

Search strategy and selection criteria

The search keywords in this systematic review were applied using Boolean operators (OR/AND). Six databases were searched for studies published between January 1990 and December 2023, using the following MeSH (Medical Subject Heading) terms and keywords: (Refractive correction OR Optical correction) OR (Single vision lenses OR Bifocal lenses OR Progressive-addition lenses OR Contact lenses) OR (Surgical methods OR Extraocular muscle surgery) OR (Botulinum toxin therapy OR drug therapy) OR (Prism therapy OR Optical correction) OR (Surgical treatment) AND Childhood esotropia (Infantile esotropia OR Congenital esotropia OR Accommodative esotropia OR Refractive esotropia OR Non-refractive accommodative esotropia OR Partially accommodative esotropia).

Eligibility and inclusion criteria

The scope of this review was restricted to scholarly articles published in peer-reviewed journals and written in English. Only studies focused on treating childhood esotropia using various management options for different types of esotropia were included. Articles that did not focus on children or did not evaluate esotropia treatments were excluded. Additionally, meeting abstracts, editorial discussions, conference papers, and studies lacking essential data collection were excluded, as illustrated in Fig. 1.

Figure 1 PRISMA flow diagram for systematic review.

Data extraction

Researchers Saif Alrasheed and Naveen Challa evaluated the titles and abstracts of all articles included in the review, using a standardized form to record information such as the first author’s name, publication year, country of study, subject characteristics (age, sample size), types of childhood esotropia, treatment methods, and the success rate of each treatment modality. In this study, success was defined as a post-treatment deviation of less than 10 prism dioptres (PD) following intervention (optical correction, extraocular muscle surgery, and botulinum toxin injection). In this systematic review, specific standards and protocols were established to manage disagreements among authors and guide the review process. These protocols fostered transparent communication, enabling authors to express differing perspectives and resolve conflicts by adhering to predefined guidelines. Any disagreements between authors were discussed and resolved based on objective criteria rather than assumptions or opinions. If a conflict remained unresolved after discussions, an external referee Abdelaziz Elmadina, serving as a neutral expert in the field, was consulted to provide insight and resolve the issue.

Risk of bias assessment

The quality of each study included in this systematic review was assessed using the evaluation tool developed by Downs & Black (1998). Additionally, Egger and Begg tests were applied to evaluate the risk of bias in the selected studies. Two review authors, Saif Alrasheed and Naveen Challa, independently conducted the bias assessment, and any discrepancies were resolved through discussion, with a third author Saeed Aljohani consulted when necessary.

Results

Study characteristics

A total of 3,605 articles were initially identified, as shown in Fig. 1. After removing duplicates, the titles of 2,457 articles were screened. Of these, 2,370 were excluded based on abstract review as they did not meet the inclusion criteria. An additional 53 articles were excluded for being conference abstracts, review articles, or book chapters, or because essential information could not be extracted, as shown in Fig. 1. The final systematic review included 34 quality-assessed studies (Mohney et al., 2011; Page et al., 2021; Downs & Black, 1998; Gerling & Arnoldi, 2013; Mohan & Sharma, 2014; Whitman, MacNeill & Hunter, 2016; Mezera, Wygnanski-Jaffe & Stolovich, 2015; Sreelatha et al., 2022; Iqbal, Hussain & Qazi, 2013; Mulvihill et al., 2000; Krishnamoorthy, 2019; Choe, Yang & Hwang, 2019; Kekunnaya, Velez & Pineles, 2013; Lambert et al., 2003; Yun-chun & Longqian, 2011; Wabulembo & Demer, 2012; Akar et al., 2013; Pensiero et al., 2021; Alshamlan et al., 2021; AlShammari, Alaam & Alfreihi, 2022; Scott et al., 1990; De Alba Campomanes, Binenbaum & Eguiarte, 2010; McNeer, Tucker & Spencer, 1997; Campos, Schiavi & Bellusci, 2000; Gursoy et al., 2012; Flores-Reyes et al., 2016; Arnoldi, 2002; Alam et al., 2023; De Alba Campomanes, Binenbaum & Eguiarte, 2010; Wan et al., 2017; Gama et al., 2020; Jiang et al., 2023; Shi et al., 2021; Bayramlar et al., 2014; Elkhawaga et al., 2022; Kim & Choi, 2017) from 14 countries, as summarized in Table 1. These studies were published between 1990 and 2023 and encompassed a total sample size of 3,877 children with a mean age of 4.72 ± 2.08 years. During the literature review and our examination of articles included in the study, we found that most studies defined treatment success as a deviation of less than 10 prism dioptres.

The reviewed studies (Mohney et al., 2011; Page et al., 2021; Downs & Black, 1998; Gerling & Arnoldi, 2013; Mohan & Sharma, 2014; Whitman, MacNeill & Hunter, 2016; Mezera, Wygnanski-Jaffe & Stolovich, 2015; Sreelatha et al., 2022; Iqbal, Hussain & Qazi, 2013; Mulvihill et al., 2000; Krishnamoorthy, 2019; Choe, Yang & Hwang, 2019; Kekunnaya, Velez & Pineles, 2013; Lambert et al., 2003; Yun-chun & Longqian, 2011) (Table 2) demonstrated that optical correction methods, including single vision lenses, bifocal lenses, progressive-addition lenses, and prismatic correction, achieved a higher success rate of 79.31% in treating childhood esotropia after an average follow-up period of 5.57 years.

Table 2 Optical correction for treatment of childhood esotropia.

Author and published year	Types of esotropia	Treatment method	Follow up (year)	Outcome measure	Effective rate	Treatment outcome and conclusion	
Mohney et al. (2011)	AET	Optical correction	9.8	<10 PD	86.5%	Most children with AET still needed glasses during their teenage years.	
Gerling & Arnoldi (2013)	AET	SVL	5	<10 PD	86%	Effectiveness in the treatment of esotropia with high AC/A.	
Mohan & Sharma (2014)	AET	Full cycloplegic hyperopic correction	13	≤10 PD	79%	Most patients with AET respond well to long-term optical correction, some developed NAET and consecutive exotropia.	
Whitman, MacNeill & Hunter (2016)	PAET	Bifocal or SVL	5.5	Stereopsis	0.00%	Bifocals has not improved stereopsis outcomes compared with SVL.	
Mezera, Wygnanski-Jaffe & Stolovich (2015)	AET	PAL	5	<10 PD	100%	PAL and bifocals were similarly useful as the initial treatment of children with AET.	
Sreelatha et al. (2022)	AET	Optical correction	5	<10 PD	63%	The majority of the children with AET have an excellent result in terms of VA and binocular functions.	
Iqbal, Hussain & Qazi (2013)	AET	Full cycloplegic correction	1	<10 PD	75%	With full cycloplegic correction most of children with AET have an excellent result in terms of ocular alignment, VA, and binocular functions.	
Mulvihill et al. (2000)	AET	Full hyperopic correction	4.5	Stereopsis	89.3%	With full hyperopic correction, most children with AET achieve excellent ocular alignment, and binocular functions.	
Krishnamoorthy (2019)	AET	Optical correction	4	<10 PD	100%	Promptly treating AET with spectacles gives the best results without any surgical intervention.	
Choe, Yang & Hwang (2019)	PAET	Prism	3	≤10PD	60.5%	Prism glasses may help treat small angle PAET.	
Kekunnaya, Velez & Pineles (2013)	PAET	Optical correction	7	<10 PD	50%	In some cases, spectacles alone can effectively correct torticollis associated with Duane syndrome.	
Lambert et al. (2003)	AET	Optical correction	4	<10 PD	91%	Children with AET often no longer need glasses during grade school.	
Yun-chun & Longqian (2011)	PAET	Prism	2	<10 PD	71.4%	Prismatic correction has shown to be effective in treating small angles of residual esotropia in PAET.	
All			5.57		79.31%		
Notes.

AET Accommodative esotropia

PAET Partial accommodative esotropia

FAET Full accommodative esotropia

SVL Single vision lenses

PAL Progressive addition lenses

Prism dioptre, 10 PD.

Extraocular muscle surgery demonstrated an effectiveness rate of 71.4% over a follow-up period of 2.89 years. Meanwhile, botulinum toxin injection showed a lower effectiveness rate of 61.24% after a follow-up period of 3.15 years, as shown in Tables 2, 3 and 4.

Table 3 Summary of surgical treatment for childhood esotropia.

Author and published year	Types of esotropia	Treatment method	Follow up (year)	Outcome measure	Effective rate	Treatment outcome and conclusion	
Wabulembo & Demer (2012)	PAET	Bilateral MR recession with pulley PF	5.7	<10 PD	95.24%	MR recession with pulley PF is effective treatment for PAET with high AC/A ratio.	
Akar et al. (2013)	PAET	Faden operations	4.8	<10 PD	78.4%	Faden surgery, with or without recession, effectively treats PAET with a high AC/A ratio.	
Pensiero et al. (2021)	EIE	Bilateral MR recession	–	<10 PD and stereopsis	64%	Traditional surgery is recommended as the initial treatment option for cases not respond to a single BT injection.	
AlShammari, Alaam & Alfreihi (2022)	PAET	Bilateral MR recession	–	0–10 PD	70.9%	BMR demonstrated a higher success rate compared to BT injection.	
De Alba Campomanes, Binenbaum & Eguiarte (2010)	IET	Bilateral MR recession	–	≤10 PD	66%	Extraocular surgery was more successful than BT in managing large angle esotropia.	
Gursoy et al. (2012)	IET	Bilateral MR recession	4	≤10 PD	77%	BT and surgical treatment showed comparable results in aligning the eyes.	
Arnoldi (2002)	PAET	Bilateral MR recession	–	<10 PD	37%	Surgical treatment for PAET leads to overcorrection.	
De Alba Campomanes, Binenbaum & Eguiarte (2010)	IET	Bilateral MR recession	2	≤10 PD	66%	Surgery was more effective than BT for treating significant esotropia.	
Wan et al. (2017)	AOET	Bilateral MR recession	1.5	≤10 PD	67%	Surgical treatment had good results in AOET.	
Shi et al. (2021)	AOET	Bilateral MR recession	3	<10 PD	75%	Surgery is an effective method for treating AOET in children.	
Bayramlar et al. (2014)	IET	Bilateral MR recession	3	≤10 PD	78%	The success rate of three horizontal muscle surgeries for IET is high in medium-term follow-up.	
Elkhawaga et al. (2022)	PAET	Bilateral fenestration MR	1	≤8PD	88%	The fenestration technique effectively reduces the angle of deviation in cases of PAET.	
Kim & Choi (2017)	NAET	Bilateral MR recession	1	≤8 PD	65.7%	Surgical treatment had good result in NAET.	
All			2.89		71.4%		
Notes.

MR Medial Rectus

PF Posterior Fixation

IET Infantile Esotropia

BT Botulinum Toxin

AOET Acute-onset esotropia

Prism dioptre 10 PD

Table 4 Summary of botulinum toxin for treatment of childhood esotropia.

Author and Published year	Types of esotropia	Treatment method	Follow up(year)	Outcome measure	Effective rate	Treatment outcome and conclusion	
Pensiero et al. (2021)	IET	Bilateral BT injection	5	<10 PD and stereopsis	36.1%	Single bilateral BT injection at age 2 is the recommended first-line treatment for EIE without a vertical component.	
Alshamlan et al. (2021)	IET	Bilateral BT injection with different dosages	4	≤10 PD	75%	Using BT in dose increments is safe, efficient, and potentially cost-effective with fewer complications.	
AlShammari, Alaam & Alfreihi (2022)	PAET	Bilateral BT injection	–	0–10 PD	53.7%	BT injection had a good result in the treatment of PAET.	
Scott et al. (1990)	ET	Bilateral BT injection	2.5	≤10 PD	63.5%	The correction frequency for cases previously operated on and un-operated cases was similar, with both groups showing 10 PD or less of correction.	
De Alba Campomanes, Binenbaum & Eguiarte (2010)	IET	Bilateral BT injection	–	≤10 PD	45%	BT is highly effective for esotropia less than 30 PD. BT could replace surgery for children with mild to moderate IE.	
McNeer, Tucker & Spencer (1997)	IET	Bilateral BT injection	–	≤10 PD	89%	BT effectively treats IE, aligning the visual axes in infants and children.	
Campos, Schiavi & Bellusci (2000)	IET	Bilateral BT injection	5.2	≤10 PD	88%	After 6 months, the angle of strabismus changed in patients who received injections. For those with hyperopic RE, it was advised to use plus-lens corrections during follow-up.	
Gursoy et al. (2012)	IET	Bilateral BT injection	4	≤10 PD	68%	BT injection had a good result in the treatment of IET.	
Flores-Reyes et al. (2016)	PAET	Bilateral BT injection	1.5	≤10 PD	71.43%	BT is an effective long-term treatment for PAET. However, it has some adverse effects, such as vertical deviation, ptosis, and diplopia.	
Alam et al. (2023)	IET	Bilateral BT injection	3	0–10 PD	51%	BT can be an alternative to surgery for children who cannot have long-lasting aesthesia or when precise measurements are not possible.	
De Alba Campomanes, Binenbaum & Eguiarte (2010)	IET	Bilateral BT injection	2	≤10 PD	45%	BT is highly effective for esotropia ranging from 30 PD to 35 PD, showing comparable success rates to surgery.	
Wan et al. (2017)	AOET	Bilateral BT injection	1.5	≤10 PD	58%	BT is as effective as surgery in treating AOET after 6 months. It also shortens general anaesthesia duration and reduces healthcare costs.	
Gama et al. (2020)	IET	Bilateral BT injection	2	<10 PD and stereopsis	21.1%	BT is suggested as an alternative but not a definitive treatment for IET, especially if the surgeon or parents are hesitant about early strabismus surgery.	
Gama et al. (2020)	NAET	Bilateral BT injection	2	<10 PD and stereopsis	60%	BT can be considered the initial treatment for NAET due to its ease, safety, and long-lasting effectiveness.	
Jiang et al. (2023)	PAE	Bilateral BT injection	7	≤10 PD, Fusion and stereopsis	80%	BT is effective for managing horizontal strabismus, especially in children with smaller angle acquired esodeviation.	
Shi et al. (2021)	AOET	Bilateral BT injection	3	<10 PD	75%	BT injections are effective for treating AOET in both adults and children, producing outcomes comparable to surgery.	
All			3.15		61.24%		
Notes.

AOET Acute-Onset Esotropia

NAET Non-accommodative Esotropia

Prism dioptre 10 PD

Publication bias

Egger and Begg tests were conducted to assess the publication bias. The outcomes showed that both tests indicated no significant bias. The results of Begg’s test (P = 0.865) and Egger’s test (P = 0.512) suggested that there is no strong evidence of publication bias in the studies that examined various treatment modalities for childhood esotropia.

Discussion

Optical correction for treatment of childhood esotropia

Single vision lenses (SVL)

Single-vision lenses are a convenient method for full hyperopic correction, offering advantages such as affordability, effectiveness, ease of fitting, straightforward prescriptions, and accessibility. Over a ten-year period, Mohney et al. (2011) utilized single-vision lenses for children with AET aged between three and seventeen, reporting a high success rate of 86.5%. The authors noted that a small number of children with AET required surgery, especially boys diagnosed at an earlier age. Additionally, Gerling & Arnoldi (2013) found that single-vision lenses were effective in treating children aged 3–7 years with AET over a follow-up period of approximately five years, particularly among those with a high accommodative convergence/accommodation (AC/A) ratio. Mohan & Sharma (2014) used full cycloplegic hyperopic correction to treat AET in children with a mean age of 4.81 ± 2.64 years, reporting a success rate of 79% over a follow-up period of 12.02 ± 2.25 years. The authors concluded that most children with AET responded well to refractive correction, although some developed non-accommodative esotropia (NAET) or consecutive exotropia. Reddy et al. (2009) indicated that refractive accommodative esotropia (RAET) is diagnosed when the deviation is eliminated or reduced to within 10 prism diopters (Δ) for both near and distance fixations with full cycloplegic hyperopic correction. Some studies (Sreelatha et al., 2022; Iqbal, Hussain & Qazi, 2013; Mulvihill et al., 2000; Krishnamoorthy, 2019; Choe, Yang & Hwang, 2019; Kekunnaya, Velez & Pineles, 2013; Lambert et al., 2003; Yun-chun & Longqian, 2011; Wabulembo & Demer, 2012; Akar et al., 2013; Pensiero et al., 2021; Alshamlan et al., 2021; AlShammari, Alaam & Alfreihi, 2022), as shown in Table 2, demonstrated a high success rate in treating children with AET using full cycloplegic hyperopic correction over follow-up periods ranging from 1 to 5 years, with excellent outcomes in terms of ocular alignment, VA, and binocular functions.

Krishnamoorthy (2019) treated children with AET aged 9 months to 4 years, treating them with hyperopic correction over a four-year follow-up period. He reported that “the deviation started to decrease after three months of using glasses and never returned”, concluding that prompt treatment of AET with spectacles yields the best outcomes without the need for surgical intervention. Additionally, Lambert et al. (2003) observed that many children with AET can discontinue wearing glasses during their grade school years, with baseline hyperopia levels serving as a strong predictor of long-term success. Optical correction showed a 50% success rate in treating young children with Duane syndrome over a 7-year period. Kekunnaya, Velez & Pineles (2013) emphasized the importance of identifying and correcting RE before surgery to minimize the risk of overcorrection. In some cases, spectacles alone can effectively correct the torticollis associated with Duane syndrome.

Bifocal lenses

Children with a high AC/A ratio and AET often achieve controlled alignment at a distance with full hyperopic correction, though a deviation may persist at near fixation. To eliminate this, bifocal lenses with additional near power are recommended to reduce the residual angle at near fixation. The most common treatment method for high AC/A AET includes bifocal lenses that correct the full cycloplegic refraction with additional power ranging from +2.00 D to +3.00 D (Whitman, MacNeill & Hunter, 2016; Ludwig et al., 2005). Whitman, MacNeill & Hunter (2016) evaluated whether treatment with bifocal glasses over a 5.5-year period, compared to single-vision lenses, improved stereopsis outcomes in individuals with high AC/A ratio accommodative esotropia. They concluded that bifocals did not improve stereopsis outcomes compared with single-vision lenses, and there is no evidence that bifocals enhance sensory outcomes in children with high AC/A ratio accommodative esotropia.

Whitman, MacNeill & Hunter (2016) noted that children may face challenges in using bifocal lenses, as some adjust their head position to depend on the bifocal segment, potentially resulting in a loss of fusional divergence and accommodative capacity. In contrast, children wearing single-vision lenses often effectively control their deviations at various distances, thereby reducing the likelihood of progression to a degree that would necessitate additional eye muscle surgery. Whitman, MacNeill & Hunter (2016) recommended that controlling esotropia to less than 10 PD at distance using full hyperopic correction is sufficient for stereopsis development, even if a significant residual esodeviation remains at near fixation. Arnoldi & Shainberg (2005) conducted a comparative study on the effects of bifocals, surgery, and single-vision lenses in children with high AC/A esotropia. The results indicated that children treated with either bifocals or single-vision lenses exhibited a 5 PD reduction in near esotropia, without a corresponding decrease in the AC/A ratio. However, children in the bifocal group were less likely to achieve emmetropization and tended to show an increase in the AC/A ratio over time. Surgery effectively reduced near deviations and eliminated the difference between distance and near deviations, though more children in the surgery group lost stereoacuity compared to those in the bifocal and single-vision lens groups. Arnoldi & Shainberg (2005) found that only a few children considered suitable for bifocal lenses achieved long-term success. Additionally, economic factors should be considered, as bifocal lenses are more expensive than single-vision lenses. Consequently, eye care professionals should consider limiting the use of bifocal lenses in future clinical practice.

Progressive addition lenses (PAL)

The primary benefit of progressive addition lenses (PAL) is their cosmetic appearance and support for natural accommodation development in children with high AC/A ratio accommodative esotropia. However, a significant drawback is the challenge of fitting young children with lenses designed for adults with presbyopia (Smith, 1985). Jacob et al. used PALs in managing children with esotropia associated with a high AC/A ratio and reported that most children experienced successful correction of their near esodeviation and achieved some level of stereopsis. Additionally, they adapted easily to wearing progressive lenses without difficulties (Jacob, Beaulieu & Brunet, 1980). Mezera, Wygnanski-Jaffe & Stolovich (2015) conducted a study on the effects of PAL and bifocals in treating esotropia with a high AC/A ratio in children with a mean age of 5.36 ± 2.68 years, over a 5-year follow-up period. They reported that PALs are highly effective in treating AET, achieving favourable sensory and motor outcomes. Similarly, both PALs and bifocals proved useful in the initial treatment of children with AET. However, economic factors should be considered, as PALs are more expensive than bifocal and single-vision lenses. Consequently, eye care specialists should take these factors into account more carefully when prescribing PALs.

Prismatic correction

Surgery is typically recommended for children with partially accommodative esotropia (PAET) if fusion cannot be achieved within 6–8 weeks of hyperopic correction or if a residual deviation greater than 10 PD remains at both near and distance with full correction. Prisms are often used as an alternative management approach for PAET (Wright & Spiegel, 2013). In a 1-year follow-up study of children with PAET, 44% of patients were able to maintain good alignment or esophoria using prism glasses alone, without requiring surgery, and none experienced a decline in stereoacuity (Han & Hwang, 2009). Additionally, in cases of consecutive esotropia following exotropia surgery, prismatic correction effectively achieved satisfactory motor alignment while preserving good stereoacuity (Lee, Yang & Hwang, 2015; Lee & Hwang, 2013).

Choe, Yang & Hwang (2019) investigated prismatic correction in managing PAET over a three-year follow-up, finding a success rate of 60.5%, and concluded that prism glasses may be beneficial for small-angle PAET with good near fusion. Similarly, Yun-chun & Longqian (2011) reported a success rate of 71.4% over a two-year follow-up in treating small residual angles of esotropia in PAET using prismatic correction. Prism can serve as an alternative for patients who cannot undergo surgery and may help control the discomfort of double vision. However, some researchers have suggested that increasing prism strength could potentially worsen the deviation.

Extraocular muscle surgery

The reviewed studies (Wabulembo & Demer, 2012; Akar et al., 2013; Pensiero et al., 2021; Alshamlan et al., 2021; AlShammari, Alaam & Alfreihi, 2022; Scott et al., 1990; de Alba Campomanes, Binenbaum & Eguiarte, 2010; McNeer, Tucker & Spencer, 1997; Campos, Schiavi & Bellusci, 2000; Gursoy et al., 2012; Flores-Reyes et al., 2016; Arnoldi, 2002; Alam et al., 2023; De Alba Campomanes, Binenbaum & Eguiarte, 2010; Wan et al., 2017; Gama et al., 2020; Jiang et al., 2023; Shi et al., 2021; Bayramlar et al., 2014; Elkhawaga et al., 2022; Kim & Choi, 2017) (Table 3) demonstrated that extraocular muscle surgery had a success rate of 71.4% in treating childhood esotropia after a follow-up period of 2.89 years. Robert & Rutstein (1998) noted that children with refractive accommodative esotropia (RAET) may not require extraocular muscle surgery; however, surgery becomes an option if a residual deviation greater than 15 PD remains after full cycloplegic hyperopic correction and poses cosmetic concerns. Traditional surgical options for children with AET associated with a high AC/A ratio include medial rectus (MR) recession with augmentation, slanted MR recession, and MR recession combined with a Faden operation or scleral posterior fixation (PF) (Wabulembo & Demer, 2012; Akar et al., 2013; Gharabaghi & Zanjani, 2006). In a study by Wabulembo & Demer (2012), the effectiveness of bilateral MR recession with a pulley PF procedure was evaluated in children with PAET. This study included children with a mean age of 4.3 ± 1.6 years, followed for 5.7 years, and reported a high success rate of 95.24%. The authors concluded that the procedure is both effective and safe, providing long-term outcomes that surpass those of other surgical approaches.

Akar et al. (2013) demonstrated a successful outcome of Faden operations for treating children with PAET over a 4.8-year follow-up. They concluded that Faden surgery, with or without muscle recession, is effective for PAET associated with a high AC/A ratio. Studies (Pensiero et al., 2021; Alshamlan et al., 2021; AlShammari, Alaam & Alfreihi, 2022; Scott et al., 1990; De Alba Campomanes, Binenbaum & Eguiarte, 2010; McNeer, Tucker & Spencer, 1997; Campos, Schiavi & Bellusci, 2000; Gursoy et al., 2012; Flores-Reyes et al., 2016; Arnoldi, 2002; Alam et al., 2023; De Alba Campomanes, Binenbaum & Eguiarte, 2010; Wan et al., 2017; Gama et al., 2020; Jiang et al., 2023; Shi et al., 2021; Bayramlar et al., 2014; Elkhawaga et al., 2022; Kim & Choi, 2017) conducted among children with various forms of esotropia, employing traditional extraocular muscle surgery or bilateral medial rectus (MR) recession, reported success rates ranging from 37% to 78%. In a study by Bayramlar et al. (2014), children aged 1–14 years with infantile esotropia (IET) who underwent bilateral MR recession over a three-year follow-up had a high success rate of 78%. The authors noted that three horizontal muscle surgeries for IET maintain high success in medium-term follow-up. In contrast, Arnoldi (2002) reported a lower success rate of 37% for bilateral MR recession in children with PAET. Arnoldi and colleagues concluded that surgical treatment for PAET can lead to overcorrection, resulting in consecutive exotropia following standard surgery. Findings from the aforementioned studies (Arnoldi, 2002; Bayramlar et al., 2014) indicated that bilateral MR recession achieved better outcomes in children with IET compared to those with PAET. Notably, the bilateral medial rectus fenestration procedure showed a high success rate of 88% in managing PAET. The authors concluded that the fenestration technique is effective in reducing the angle of deviation in PAET cases (Elkhawaga et al., 2022).

Botulinum toxin injection (BT)

The reviewed studies (Pensiero et al., 2021; Alshamlan et al., 2021; AlShammari, Alaam & Alfreihi, 2022; Scott et al., 1990; de Alba Campomanes, Binenbaum & Eguiarte, 2010; McNeer, Tucker & Spencer, 1997; Campos, Schiavi & Bellusci, 2000; Gursoy et al., 2012; Flores-Reyes et al., 2016; Arnoldi, 2002; Alam et al., 2023; De Alba Campomanes, Binenbaum & Eguiarte, 2010; Wan et al., 2017; Gama et al., 2020; Jiang et al., 2023; Shi et al., 2021) (Table 4) showed that bilateral botulinum toxin (BT) injections at various dosages achieved a 61.24% success rate in treating childhood esotropia over a follow-up period of 3.15 years. Injecting BT into the extraocular muscles induces temporary paralysis and overcorrection of the strabismus, which promotes shortening of the opposing muscle. Histological studies have demonstrated sarcomere density changes that enhance long-term ocular alignment (Kowal, Wong & Yahalom, 2007). BT type A has proven effective for small to moderate-angle esotropia, acute or chronic fourth and sixth nerve palsy, childhood strabismus, and thyroid eye disease. Pensiero et al. (2021) reported a lower success rate of 36.1% following BT chemo-denervation for children with IET over a one-year follow-up. They concluded that a single bilateral BT injection at age 2 is a recommended first-line treatment for IET without a vertical component. These studies (McNeer, Tucker & Spencer, 1997; Campos, Schiavi & Bellusci, 2000; Gursoy et al., 2012) conducted on children with IET using bilateral botulinum toxin (BT) injections reported high success rates ranging from 68% to 89%. McNeer, Tucker & Spencer (1997) found that BT injections effectively align the visual axes in infants and children with IET. In contrast, other studies (De Alba Campomanes, Binenbaum & Eguiarte, 2010; Gama et al., 2020) reported lower success rates of 21.1% and 45% for bilateral BT injections in treating IET. Bilateral BT injection is often recommended as an alternative rather than a definitive treatment for IET, particularly when the surgeon or parents are hesitant about early strabismus surgery.

Flores-Reyes et al. (2016) demonstrated a significant reduction in deviation following BT injection, with a success rate of 71.43%. The optimal effectiveness of the toxin spanned 6 to 12 months; however, the study’s follow-up period was limited to 1.5 years, lacking data on long-term outcomes. Flores concluded that BT is an effective long-term treatment for PAET but noted potential adverse effects, including vertical deviation, ptosis, and diplopia. Wan et al. (2017) reported a success rate of 58% for bilateral BT injections in treating children with acute-onset esotropia (AOCET) over a 1.5-year follow-up. Wan suggested that BT is as effective as surgery in treating AOCET after six months, with the added benefits of reduced anaesthesia time and healthcare costs. Additionally, Shi et al. (2021) reported a high success rate of 75% for bilateral BT injections in treating AOCET in children over three years. Shi and colleagues concluded that BT injections are effective for managing AOCET in both adults and children, providing results comparable to surgical intervention.

Patching and vision therapy

Strabismic amblyopia is commonly associated with childhood esotropia, particularly in those with partially accommodative esotropia (Mezera, Wygnanski-Jaffe & Stolovich, 2015). Clinically, eye care providers must carefully assess for amblyopia in children with esotropia. When accommodative esotropia is accompanied by amblyopia, treatment for amblyopia should be initiated concurrently with the management of the esotropia. The treatment of childhood amblyopia associated with esotropia typically involves a combination of refractive correction, occlusion therapy, and vision therapy. Interventions such as dichoptic and perceptual training have shown effectiveness in managing strabismic amblyopia (Alrasheed & Aldakhil, 2024). In this review, the most commonly employed approach for treating children with accommodative esotropia and amblyopia was the correction of cycloplegic refractive error, followed by occlusion therapy.

Limitations of study

Our systematic review had some limitations: firstly, some of the reviewed studies were retrospective in nature. Secondly, there was no consistent standard for defining effectiveness rates, or for setting inclusion and exclusion criteria. Thirdly, the studies varied in sample size, with some being small and others large, which may have increased the risk of random error. Additionally, the present review did not include amblyopia or the assessment of visual acuity, nor did it address the use of patching and vision therapy, which are common treatment modalities for childhood esotropia. We recommend that future research should prioritize more randomized controlled trials with larger sample sizes. These trials should adhere to standardized inclusion and exclusion criteria and use comparable treatment protocols. Additionally, implementing long-term follow-up periods will be crucial to gather comprehensive data on the various treatments for childhood esotropia. Despite these limitations, this review provides valuable and updated information on the effectiveness of different treatment modalities for improving ocular alignment in childhood esotropia.

Conclusions

Substantial evidence indicates that full cycloplegic hyperopic correction is the most effective first-line treatment for childhood AET. Single-vision lenses and progressive addition lenses are recommended for children with AET associated with a high AC/A ratio. However, the reviewed studies indicated that if the deviation remains greater than 15 PD after optical correction, surgery is considered necessary for both functional and cosmetic reasons. Prismatic correction has shown high success rates in managing small residual esotropia in PAET associated with hyperopia. For infantile esotropia without a vertical component, bilateral botulinum toxin injection by age two is an effective first-line treatment, while bilateral medial rectus recession is preferred for all other cases and those unresponsive to initial treatments. Timely surgical intervention enhances sensory outcomes in infantile esotropia; however, no single surgical technique has a distinct advantage. Evidence also supports the effectiveness of BT injections in treating acute onset esotropia in children, with outcomes comparable to those of surgery.

Supplemental Information

Supplemental Information 1 PRISMA checklist

The authors would like to express our sincere gratitude to the external referee, Dr. Abdelaziz Elmadina, for his valuable insights and constructive feedback, which greatly contributed to the improvement of this review.

Additional Information and Declarations

Competing Interests

Author Contributions

Data Availability

The authors declare there are no competing interests.

Saif Hassan Alrasheed conceived and designed the experiments, performed the experiments, analyzed the data, prepared figures and/or tables, authored or reviewed drafts of the article, and approved the final draft.

Naveen Kumar Challa conceived and designed the experiments, performed the experiments, analyzed the data, prepared figures and/or tables, authored or reviewed drafts of the article, and approved the final draft.

Saeed Aljohani conceived and designed the experiments, performed the experiments, prepared figures and/or tables, authored or reviewed drafts of the article, and approved the final draft.

Nawaf M. Almutairi conceived and designed the experiments, performed the experiments, prepared figures and/or tables, authored or reviewed drafts of the article, and approved the final draft.

Mohammed M. Alnawmasi conceived and designed the experiments, performed the experiments, prepared figures and/or tables, authored or reviewed drafts of the article, and approved the final draft.

The following information was supplied regarding data availability:

This is a systematic review/meta-analysis.

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
