# Peer review of "Effectiveness of treatment modalities for childhood esotropia: a systematic review"

_PeerJ, doi:10.7717/peerj.19584_

## Round 0.1 · original submission · Major Revisions

Please respond to all comments in details

Reviewer 1 ·

Basic reporting

See below

Experimental design

See below

Validity of the findings

See below

Additional comments

This systematic review looked into the various treatment modalities of the various types of childhood esotropia. The authors reported that optical correct has the highest effectiveness rate, and concluded that full cycloplegic hyperopic correction is the most effective treatment for accommodative esotropia, whereas prismatic correction and bilateral botulinum toxin injection are appropriate for acute-onset esotropia. The authors have summarized the studies in terms of the proportions of children who have been successfully treated with each treatment modality. Although treatment success is defined as <10 PD deviation in the methods section, they have not explained the rationale, and they also included studies with various outcome measures. For example, one study was reported as poor treatment success based on stricter criteria, i.e. stereopsis. In this instance, the authors should have a column in the table showing clearly the outcome measure for each study. Interestingly, the authors did not mention about amblyopia and visual acuity, or the use of patching and vision therapy even though these are common treatment modalities as well. Minor comments: In the results section, the authors did not state the type of bifocal lenses. Line 138 “proven effectiveness”. Line 356: Why is surgery necessary?

Reviewer 2 ·

Basic reporting

English Language Usage

The article is written in understandable English; however, some sentences contain grammatical errors. A thorough English language review is recommended to ensure clarity and correctness. Improving word choice and sentence structure in some sections would enhance readability.

References

The references cited in the article are sufficient and relevant to the content.
Introduction

The background and introduction provide a good foundation for the study. However, some corrections are needed to ensure clearer wording and flow. Refining the phrasing and organization in these sections could make the content more accessible and engaging to readers.
Line 70 AET usually occurs between 2-4 years old and not during the first year of life. Please amend the statement.
Line 70-72 I suggest refining the phrasing.
Line 87 Accommodation has been identified as a significant factor…….I would suggest to replace the factor with contributor to make the sentences more precise.
Line 95-96 I suggest replacing the phrase “enhancing ocular misalignment” with improving ocular alignment. The aim of strabismus management is to ensure patient has better ocular alignment.
Line 97-99 I would also suggest rephrasing this sentence for better clarity.

Experimental design

Methodology
The raw data provided is sufficient and supports the findings presented in the article.
The figures and tables are adequate and relevant to the content of the article. However, Figure 1 requires modification to align with the PRISMA template. The author omitted one element, which is Eligibility. Ensuring compliance with the template will improve the figure's accuracy.
I suggest including the month when the authors conducted the literature search in the text.
Line 134-35 "Articles that did not focus on children or ……..”. I suggest deleting this statement as it conveys the same meaning as the previous sentence.

Results
The results section is too brief and does not comprehensively present the findings of the study. In my opinion, the authors appear to have integrated the results into the discussion section. Lines 183–186: These statements outline the findings based on the studies evaluated, rather than contributing to a discussion. This content should be moved to the results section.
Lines 190–205: Similarly, this portion contains findings derived from the evaluation of 34 articles. These should also be moved to the results section.
Similarly, other parts of the discussion section also represent findings rather than discussion. I suggest these sections should be carefully reviewed and moved to the results section.
I would suggest including the type of studies (cross-sectional, cohort or retrospective studies) in this section and in Table 1.

Validity of the findings

The authors have highlighted the importance and potential benefits of the study, providing clear insights into its relevance and application.
The conclusion answers to the research question, reinforcing the study's objectives and outcomes.

Additional comments

The article has a strong foundation with sufficient references, relevant figures and tables, and adequate raw data. Addressing the suggested revisions will enhance the quality and clarity of the manuscript.

Reviewer 3 ·

Basic reporting

Saif Alrasheed et al. have conducted an extensive review of treatment outcomes in strabismus patients, specifically focusing on esotropic patients. The article is clear and technically accurate. The authors made an effort to synthesize all research on treatments to develop a more effective treatment protocol for patients.

1. The authors could explain why treatments for other types of strabismus, such as exotropia, were not included in the investigation.

2. Are there any differences in treatment outcomes between esotropic patients with a family history and those with infantile esotropia due to disorders or acquired strabismus?

3. A final table with the most effective (mean percentage) treatment options for different types of strabismus can be included.

4. Line 53 "general, strabismus affects approximately 4% of the population." citation is recommended

5. Line 134 "management options for different types of esotropia were included." Acquired strabismus was included in the study? expand or explain.

6. Line 171 "The reviewed studies indicated that optical correction had a higher effectiveness rate of 79.31%" This mean doesn't include whitman study?

7. Table 1 is missing SD on Age for some cases, include if possible.

8. Were there different outcomes depending on the study region?

Experimental design

The research question is well-defined and gives us an idea of treatment outcomes.

Validity of the findings

no comment

Annotated reviews are not available for download in order to protect the identity of reviewers who chose to remain anonymous.

---

## Round 0.2 · Minor Revisions

Thank you for your thorough responses to the reviewer comments. While the conclusions are solidly supported, there remains a concern that the authors have not really discussed patching and vision therapy despite acknowledging in the paper that these are common treatments for childhood esotropia. Please add a few more sentences on this topic covering how often these treatments are used (if known), under what circumstances these treatments might be considered, and the success rate of these treatments (if known).

Reviewer 1 ·

Basic reporting

Thank you for the revised manuscript. The authors have adequately addressed the reviewer comments.

Experimental design

No comment

Validity of the findings

No comment

Reviewer 3 ·

Basic reporting

The authors have addressed each point I commented on in the manuscript. They have presented an extensive review aimed at finding better outcomes in strabismus treatment, more specifically for esotropia. The findings are significant and have important implications for the field, allowing caregivers, based on the literature, to provide better treatment options according to the strabismus type. Overall, the manuscript is clear, concise, and well-described. The introduction is relevant to the study, where the authors discuss different types of esotropia. Overall, the results are clear and compelling. The authors make a systematic contribution to the research literature in this area of investigation. This manuscript has important implications for the field, particularly in understanding the possible outcomes for different types of esotropia and potentially unifying the use of treatments depending on the esotropia case.

Experimental design

no comment

Validity of the findings

no comment

Additional comments

no comment

---

## Round 0.3 · accepted · Accept

Thank you again for your responses to the reviewer and editor comments.